# Analysis of Self-Concept in Adolescents before and during COVID-19 Lockdown: Differences by Gender and Sports Activity

**Gabriel González-Valero [1], Félix Zurita-Ortega [1], David Lindell-Postigo [2], Javier Conde-Pipó [1], Wilhelm Robert Grosz [3] and Georgian Badicu [3,*]**

[1] Department of Didactics of Musical, Plastic and Corporal Expression, University of Granada, Campus de Cartuja, s/n 18071 Granada, Spain; ggvalero@ugr.es (G.G.-V.); felixzo@ugr.es (F.Z.-O.); javiconde@correo.ugr.es (J.C.-P.)

[2] Novaschool Sunland International, Estación de Cártama-Málaga, Carretera de Cártama Estación a Pizarra, s/n 29580 Cártama, Spain; dlindell@novaschool.es

[3] Department of Physical Education and Special Motricity, Faculty of Physical Education and Mountain Sports, Transilvania University of Brasov, 500068 Brasov, Romania; wilhelm.grosz@unitbv.ro

[*] Correspondence: georgian.badicu@unitbv.ro; Tel.: +40-769-219-271

**Abstract:** An appeal has been issued to the scientific community to investigate physical, mental and emotional states, and pro-social behaviours during the COVID-19 pandemic. Hence, this study aims to investigate adolescents' self-concept before and during a lockdown period in relation to gender and type/amount of physical activity or sports. The pre-lockdown sample of 366 adolescents were aged 13–17 years (M = 15.51 ± 0.65), of whom 192 (52.5%) were females and 174 (47.5%) were males. During the lockdown, the age range of the sample was 13–17 years (M = 14.57 ± 1.47), of whom 82 (60.3%) were females, and 54 (39.7%) were males. The Form-5 Self-concept Questionnaire (AF-5) was used to measure adolescents' self-concept. There was a reduction in adolescents' overall self-concept during the COVID-19 pandemic, which was positively associated with emotional well-being, with family and peers being essential factors in the development of an appropriate self-concept. Furthermore, girls' self-concept, especially academic self-concept, was higher than that of boys during the lockdown. However, both physical and emotional self-concept were higher for boys than girls before the COVID-19 lockdown, although no differences were found during the lockdown. The findings reveal that physical activity was positively correlated to self-concept before and during the COVID-19 lockdown.

**Keywords:** self-concept; physical activities; lockdown; COVID-19; adolescents

## 1. Introduction

Every country negatively affected by a disaster or a pandemic considers adolescents, the elderly and disabled people to be the main at-risk population groups [1–3]. The coronavirus SARS-CoV-2 infection, which leads to the disease called COVID-19 [4], caused an international state of health emergency and global pandemic [5]. This disease features respiratory infections, which directly affect the elderly [4,6]. However, adolescents are indirectly affected due to social distancing, educational and recreational measures adopted by the government [7,8]. The pandemic has had economic impacts and caused social disruptions [9,10].

Furthermore, COVID-19 globally jeopardises people's mental health since it increases stress, anxiety, depression and negative social behaviour [9–11]. Therefore, adolescents are in danger of experiencing negative consequences for both mental and physical health as a result of pandemics

and disasters [2,12]. Likewise, COVID-19 is affecting aspects of daily life, such as educational, social and leisure activities, which present both familial and emotional challenges [7,8,13,14]. Moreover, psychological needs, such as self-fulfilment, self-esteem and affective relationships, take on greater importance once both physiological and security requirements are met [15].

It should be noted that self-concept, which is understood as one's perception about oneself or the general opinion about self-esteem [16], may buffer people's psychological distress [17,18]. Hence, self-concept is especially important in adolescence, since everyday emotions and feelings are essential in personal development, which is subjective and changes according to external factors and new contexts of life [19]. As a matter of fact, self-concept represents a protective factor against disruptive behaviour, enhancing both mental health and positive peer relationships [20]. Consequently, the psychological construction of a positive self-concept in students during the school years produces successful socio-emotional situations and educational settings [21].

The literature has shown gender differences in well-being and self-concept. Females' well-being and self-concept are related to life satisfaction and happiness, whereas males' well-being and self-concept are related to feelings of achievement [22,23]. The effect of the COVID-19 pandemic has not been studied and no conclusive results have therefore been drawn. The need to study the association between self-concept and physical activity during this period lies in the importance of children's development, with the physical practice being a means of improving mental processes and socialisation in children [19]. Adolescents who habitually participate in physical and sports activities have a better self-concept, which is the social motor that drives better academic performance and helps their relations with peers [24].

Hence, this construct should be considered in youths and their relatives who usually experience post-traumatic symptoms under pandemic or emergency situations [9,11,25], which may trigger negative mental health consequences, disruptive behaviour and a low self-concept [26–29]. As a consequence of the coronavirus pandemic, an appeal has been issued to the scientific community to pay attention to and analyse the physical, mental and emotional conditions as well as people's pro-social behaviour [13,30,31]. In relation to the study problem, the following research questions were suggested: (a) are there differences in adolescents' levels of self-concept before and during COVID-19? (b) are there differences between boys and girls? (c) does being physically active or inactive influence adolescents' self-concept before and during COVID-19? Therefore, this study aims to examine the level of adolescents' self-concept before lockdown and during lockdown as regards gender and physical activity, for boys and girls, and for those who are physically active and those who are not.

## 2. Materials and Methods

### 2.1. Design and Participants

This study compared the self-concept of two groups of adolescents, one pre-lockdown and one during lockdown. Convenience sampling was used to select participants, in which adolescents were asked to participate before and during the lockdown, so different samples were evaluated at two different times. As regards this selection criterion, 72.9% ($n$ = 366) of participants were assessed before the COVID-19 lockdown and 27.1% ($n$ = 136) of participants were assessed during that period. Both groups were equivalent in all respects, except for the lockdown situation. The age range of the sample before the lockdown was 13–17 years (M = 15.51 ± 0.65), while females accounted for 52.5% ($n$ = 192), males accounted for 47.5% ($n$ = 174). During the lockdown, the age range of the sample was 13–17 years (M = 14.57 ± 1.47); females constituted 60.3% ($n$ = 82) of the total, whereas males accounted for 39.7% ($n$ = 54). In order to obtain a representative sample (error at 0.05; C.I. = 95%), stratification and proportionality techniques were used when establishing the groups.

## 2.2. Instruments and Variables

Ad-hoc questionnaire. This instrument was created by the researchers and was used to collect socio-demographic and physical and sports data. Thus, the data collected were participants' gender and age, moment (in lockdown or not), and type of physical or sport activities practised according to the adapted classification of Castro-Sánchez et al. [32]. This classification involves the following categorisation: 'None', 'Non-contact individual sports' (NCIS), 'Contact individual sports' (CIS), 'Non-contact team sports' (NCTS) and 'Contact team sports' (CTS). Furthermore, based on this classification, adolescents were classified as "physically active" and "physically inactive".

Form-5 Self-concept Questionnaire (AF-5). This questionnaire assesses one's perception about one's self-concept, which is based on the theory stated by Shavelson, Hubner and Stanton [33], and was created and validated to Spanish by García and Musitu [34]. It consists of 30 items, which use five-points Likert responses ranging from 'Never' to 'Always'. Item summation allows us to establish a general measurement of this construct, as well as to group self-concept into five dimensions: academic self-concept (A-SC; items 1, 6, 11, 16, 21 and 26), social self-concept (S-SC; items 2, 7, 12, 17, 22 and 27), emotional self-concept (E-SC; items 3, 8, 13, 18, 23 and 28), family self-concept (F-SC; items 4, 9, 14, 19, 24 and 29) and physical self-concept (P-SC; items 5, 10, 15, 20, 25 and 30). Cronbach's alpha in this study ($\alpha = 0.809$) was similar to García and Musitu [34] research ($\alpha = 0.810$). Reliabilities of each dimension of self-concept were as follows: A-SC ($\alpha = 0.853$), S-SC ($\alpha = 0.784$), E-SC ($\alpha = 0.756$), F-SC ($\alpha = 0.706$) and P-SC ($\alpha = 0.801$).

## 2.3. Procedure

Firstly, researchers explored the range of ways to contact the population. Adolescents evaluated before the lockdown were asked thenceforth to participate in this investigation via an informative letter, which was created by the Body language department of the University of Granada and delivered through their schools. A meeting with principals was arranged afterwards, in which researchers handed over to principals some hard copies of evaluation instruments and the informative letter that needed to be delivered to adolescents' families. That letter detailed the objectives and the nature of the research, explained the voluntariness of the participation and requested informed consent. Data collection was conducted in school time during Physical Education lessons in the presence of researchers and teachers, in order to solve any doubt and to ensure a correct completion, not occurring any incidence during the process. The procedure was similar with participants evaluated during the lockdown, although protocols were done online, questionnaires were distributed via social media, and contact with families was done through schools' communication channels (e.g., email, blogs or *Telegram*). Anonymity was ensured in both processes, and researchers also certified that data would be used for scientific purposes. The study was conducted in full compliance with the principles expressed in the Declaration of Helsinki and was approved by the Scientific Ethical Committee of the research team's university (1230/CEIH/2020). Lastly, researchers had to invalidate 37 questionnaires due to incorrect completion.

## 2.4. Data Analysis

Descriptive analysis for variables in this study was performed, calculating mean values (M), standard deviation (SD) and frequencies (%). Normality and homogeneity of variance for every variable were analysed through the Kolmogorov–Smirnov test. Researchers performed the independent Student's t-test to estimate differences among variables and performed the Bonferroni post-hoc test to determine the one-way variance (ANOVA) with one group, determined by Pearson's chi-squared test. Pearson bivariate correlation was calculated to establish an association among mean values, calculating a significance level of $p \leq 0.05$ and $p \leq 0.01$. Lastly, the magnitude of difference in effect size (ES) was obtained with Cohen's *d* index [35], which is interpreted as null (0–0.19), small (0.20–0.49), medium (0.50–0.7), and large ($\geq 0.80$) [36]. Data were analysed using IBM SPSS® version 25.0 (IBM Corp,

Armonk, NY, USA). GraphPad Prism 8 (GraphPad Prism Software Inc., San Diego, CA, USA) was used to produce figures. Lastly, 37 questionnaires were invalidated for incorrect completion, of which 26 were from individuals before the lockdown and 11 were during the COVID-19 lockdown.

## 3. Results

Table 1 presents values of self-concept regarding lockdown during the COVID-19 pandemic ($p \leq 0.05$). The highest values of total self-concept were before the lockdown (M = 3.47 ± 0.52; $d$ = 0.253), presenting a positive association with emotional self-concept ($r$ = 0.375**). This association was not present during the lockdown ($r$ = 0.090). The social self-concept (M = 3.70 ± 0.75; $d$ = 0.401) was influenced by its correlation with emotional self-concept ($r$ = 0.200**). The emotional self-concept (M = 3.02 ± 0.78; $d$ = 0.482) had a positive association with family self-concept ($r$ = 0.143**) in both moments. Likewise, there existed a negative association between physical and emotional dimensions of self-concept during the lockdown ($r$ = −0.424**). Furthermore, while individuals evaluated before the lockdown presented higher values in family self-concept (M = 4.02 ± 0.83; $d$ = 0.643), during the lockdown presented them in the academic dimension (M = 3.85 ± 0.70; $d$ = 0.705).

**Table 1.** Self-concept before and duringCOVID-19 pandemic.

| | | | | | | | | | Levene | | T-Test | | ES | IC 95% |
|---|---|---|---|---|---|---|---|---|---|---|---|---|---|---|
| Variables | Lockdown | M | SD | (2) | (3) | (4) | (5) | (6) | F | Sig. | T | Sig. | (d) | |
| G-SC (1) | Yes | 3.35 | 0.32 | 0.744 ** | 0.695 ** | 0.090 | 0.591 ** | 0.637 ** | 24.222 | 0.000 | −3.003 | 0.003 | 0.253 | [0.056; 0.450] |
| | No | 3.47 | 0.52 | 0.612 ** | 0.754 ** | 0.375 ** | 0.749 ** | 0.717 ** | | | | | | |
| A-SC (2) | Yes | 3.85 | 0.70 | 1 | 0.499 ** | −0.210 * | 0.266 ** | 0.413 ** | 9.695 | 0.002 | 7.836 | 0.000 | 0.705 | [0.504; 0.907] |
| | No | 3.25 | 0.90 | 1 | 0.247 ** | −0.125 * | 0.371 ** | 0.378 ** | | | | | | |
| S-SC (3) | Yes | 3.43 | 0.40 | | 1 | −0.130 | 0.337 ** | 0.398 ** | 58.009 | 0.000 | −5.149 | 0.000 | 0.401 | [0.202; 0.599] |
| | No | 3.70 | 0.75 | | 1 | 0.200 ** | 0.499 ** | 0.523 ** | | | | | | |
| E-SC (4) | Yes | 2.66 | 0.65 | | | 1 | 0.027 | −0.424 ** | 7.036 | 0.008 | −5.239 | 0.000 | 0.482 | [0.283; 0.681] |
| | No | 3.02 | 0.78 | | | 1 | 0.143 ** | 0.046 | | | | | | |
| F-SC (5) | Yes | 3.54 | 0.45 | | | | 1 | 0.214 * | 71.469 | 0.000 | −8.172 | 0.000 | 0.643 | [0.442; 0.844] |
| | No | 4.02 | 0.83 | | | | 1 | 0.369 ** | | | | | | |
| P-SC (6) | Yes | 3.29 | 0.76 | | | | | 1 | 0.196 | 0.658 | −0.909 | 0.364 | 0.090 | [−0.107; 0.287] |
| | No | 3.36 | 0.78 | | | | | 1 | | | | | | |

Note 1. General self-concept (G-SC); academic self-concept (A-SC); social self-concept (S-SC); emotional self-concept (E-SC); family self-concept (F-SC); physical self-concept (P-SC). Note 2. Significative correlation at 0.05 (*); significative correlation at 0.01 (**).

Figure 1 shows the relation between self-conceptandthe lockdown. The general self-concept before the lockdown was higher than during this period. The same happened with the S-SC, E-SC and F-SC. In contrast, the A-SC was higher during the lockdown period. No statistically significant results were found for the P-SC.

Table 2 presents differences between gender and self-concept before the COVID-19 lockdown. Girls had a greater academic self-concept (M = 3.39 ± 0.90; $d$ = 0.339) with a highly significant difference ($p$ = 0.001). However, among the boys, a negative association was shown between that dimension and emotional self-concept ($r$ = −0.287**). There was a significant difference ($p \leq 0.05$) in some components before the lockdown. Boys had specifically higher values in the P-SC (M = 3.44 ± 0.77; $d$ = 0.320) and E-SC (M = 3.14 ± 0.73; $d$ = 0.298). However, girls' E-SC was positively correlated with the F-SC ($r$ = 0.154*).

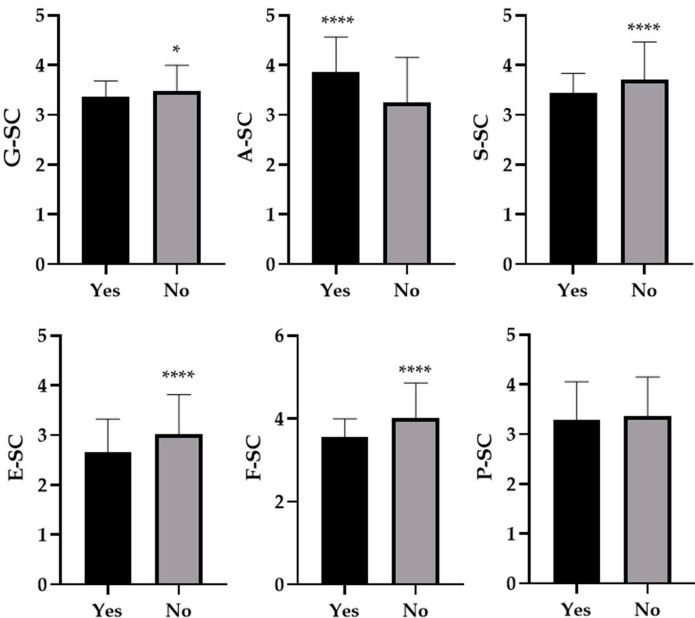

**Figure 1.** Self-concept and dimensions according to the period of lockdown. Note 1. Self-concept (SC); general self-concept (G-SC); academic self-concept (A-SC); social self-concept (S-SC); emotional self-concept (E-SC); family self-concept (F-SC); physical self-concept (P-SC). Note 2. Lockdown (Yes); nolockdown (No). Note 3. $p \leq 0.05$ (*); $p \leq 0.0001$ (****).

**Table 2.** Gender-based differences in self-concept before COVID-19 lockdown.

| Lockdown | Variables | Sex | M | SD | (2) | (3) | (4) | (5) | (6) | Levene F | Levene Sig. | T-Test T | T-Test Sig. | ES (d) | IC 95% |
|---|---|---|---|---|---|---|---|---|---|---|---|---|---|---|---|
| | G-SC (1) | B | 3.48 | 0.48 | 0.526 ** | 0.768 ** | 0.288 ** | 0.730 ** | 0.734 ** | 2.598 | 0.108 | 0.379 | 0.705 | 0.039 | [−0.167; 0.244] |
| | | G | 3.46 | 0.55 | 0.702 ** | 0.747 ** | 0.439 ** | 0.763 ** | 0.709 ** | | | | | | |
| | A-SC (2) | B | 3.09 | 0.87 | 1 | 0.165 * | −0.287 ** | 0.220 ** | 0.386 ** | 0.026 | 0.873 | −3.240 | 0.001 | 0.339 | [0.132; 0.545] |
| | | G | 3.39 | 0.90 | 1 | 0.342 ** | 0.044 | 0.493 ** | 0.413 ** | | | | | | |
| SC before lockdown | S-SC (3) | B | 3.74 | 0.76 | | 1 | 0.196 ** | 0.497 ** | 0.523 ** | 0.032 | 0.858 | 0.851 | 0.395 | 0.093 | [−0.113; 0.298] |
| | | G | 3.67 | 0.75 | | 1 | 0.197 ** | 0.505 ** | 0.519 ** | | | | | | |
| | E-SC (4) | B | 3.14 | 0.73 | | | 1 | 0.140 | −0.065 | 2.199 | 0.139 | 2.830 | 0.005 | 0.298 | [0.091; 0.504] |
| | | G | 2.91 | 0.81 | | | 1 | 0.154 * | 0.112 | | | | | | |
| | F-SC (5) | B | 4.00 | 0.79 | | | | 1 | 0.376 ** | 3.381 | 0.067 | −0.451 | 0.652 | 0.048 | [−0.157; 0.253] |
| | | G | 4.04 | 0.88 | | | | 1 | 0.371 ** | | | | | | |
| | P-SC (6) | B | 3.44 | 0.77 | | | | | 1 | 0.024 | 0.877 | 1.789 | 0.004 | 0.320 | [0.114; 0.527] |
| | | G | 3.19 | 0.79 | | | | | 1 | | | | | | |

Note 1. Boy (B); girl(G). Note 2. Self-concept (SC); general self-concept (G-SC); academic self-concept (A-SC); social self-concept (S-SC); emotional self-concept (E-SC); family self-concept (F-SC); physical self-concept (P-SC). Note 3. Significative correlation at 0.05 (*); significative correlation at 0.01 (**).

Figure 2 shows the relationship between self-concept and gender-based differences before lockdown. Girls had greater levels ofA-SC, although boys had higher E-SC and P-SC. No statistically significant results ($p \geq 0.05$) were detected for the G-SC, S-SC and F-SC.

Table 3 presents differences according to gender regarding self-concept during the COVID-19 lockdown. There were also significant differences ($p \leq 0.05$) during lockdown. Females had greater values in A-SC (M = 3.97 ± 0.70; $d = 0.421$), while for males, A-SC had a negative association with E-SC ($r = -0.281$*) and a positive correlation with F-SC ($r = 0.414$**). Significant differences were shown in G-SC ($p = 0.027$), where girls had greater values than boys (M = 3.38 ± 0.31; $d = 0.346$).

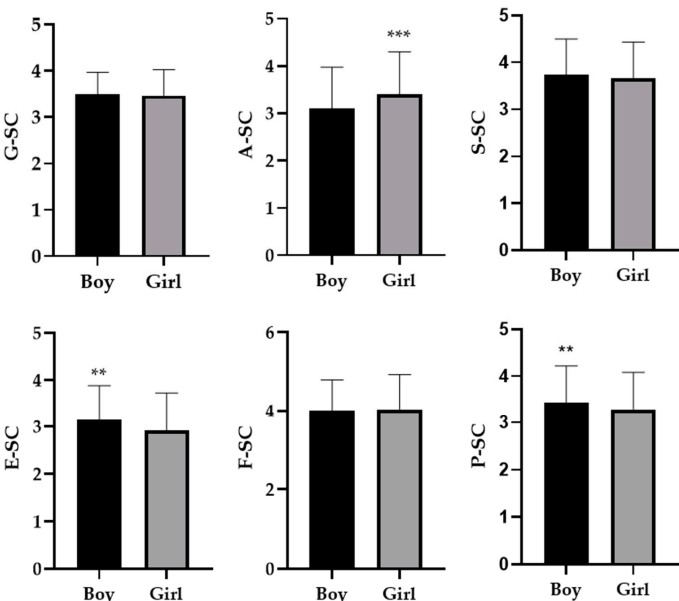

**Figure 2.** Gender-based differences in self-concept before COVID-19 lockdown. Note 1. Self-concept (SC); general self-concept (G-SC); academic self-concept (A-SC); social self-concept (S-SC); emotional self-concept (E-SC); family self-concept (F-SC); physical self-concept (P-SC). Note 2. $p \leq 0.01$ (**); $p \leq 0.001$ (***).

**Table 3.** Gender-based differences in self-concept during COVID-19 lockdown.

| Lockdown | Variables | Sex | M | SD | (2) | (3) | (4) | (5) | (6) | Levene F | Levene Sig. | T-Test T | T-Test Sig. | ES (d) | IC 95% |
|---|---|---|---|---|---|---|---|---|---|---|---|---|---|---|---|
| SC during lockdown | G-SC (1) | M | 3.27 | 0.33 | 0.837 ** | 0.706 ** | −0.024 | 0.614 ** | 0.710 ** | 0.520 | 0.729 | −2.092 | 0.027 | 0.346 | [0.001; 0.692] |
| | | F | 3.38 | 0.31 | 0.682 ** | 0.695 ** | 0.170 | 0.576 ** | 0.582 ** | | | | | | |
| | A-SC (2) | M | 3.68 | 0.67 | 1 | 0.596 ** | −0.281 * | 0.414 ** | 0.602 ** | 0.335 | 0.564 | −2.375 | 0.019 | 0.421 | [0.074; 0.768] |
| | | F | 3.97 | 0.70 | 1 | 0.456 ** | −0.170 | 0.175 | 0.289 ** | | | | | | |
| | S-SC (3) | M | 3.43 | 0.39 | | 1 | −0.231 | 0.320 * | 0.484 ** | 0.014 | 0.905 | −0.040 | 0.968 | 0.001 | [−0.343; 0.343] |
| | | F | 3.43 | 0.41 | | 1 | −0.072 | 0.349 ** | 0.339 ** | | | | | | |
| | E-SC (4) | M | 2.67 | 0.63 | | | 1 | 0.009 | −0.484 ** | 0.472 | 0.493 | 0.185 | 0.853 | 0.031 | [−0.313; 0.374] |
| | | F | 2.65 | 0.67 | | | 1 | 0.038 | −0.387 ** | | | | | | |
| | F-SC (5) | M | 3.54 | 0.46 | | | | 1 | 0.144 | 0.147 | 0.702 | −0.109 | 0.913 | 0.001 | [−0.343; 0.343] |
| | | F | 3.54 | 0.44 | | | | 1 | 0.223 * | | | | | | |
| | P-SC (6) | M | 3.27 | 0.81 | | | | | 1 | 0.013 | 0.908 | −0.225 | 0.822 | 0.040 | [−0.304; 0.383] |
| | | F | 3.30 | 0.72 | | | | | 1 | | | | | | |

Note 1. Male/Boy (M); Female/Girl (F). Note 2. Self-concept (SC); general self-concept (G-SC); academic self-concept (A-SC); social self-concept (S-SC); emotional self-concept (E-SC); family self-concept (F-SC); physical self-concept (P-SC). Note 3. Significative correlation at 0.05 (*); significative correlation at 0.01 (**).

Figure 3 shows the relationships between self-concept and gender-based differences during lockdown. Especially the girls had a greater G-SC and A-SC than boys. No statistically significant results ($p \geq 0.05$) were found for the other dimensions.

Table 4 presents the association between practising physical activities and self-concept before the lockdown ($p \leq 0.05$). Individuals who were not involved in any physical activity had the lowest values of G-SC (M = 3.30 ± 0.49) compared to NCIS (M = 3.58 ± 0.52; $d$ = 0.554) and STS participants (M = 3.59 ± 0.53; $d$ = 0.574). Individuals who practiced NCIS had the highest values in A-SC (M = 3.47 ± 0.87; $d$ = 0.905) compared to individuals who did not practice any physical activity (M = 2.73 ± 0.76). The practice of NCIS or CTS was associated to higher values of S-SC (M = 3.83 ± 0.68; $d$ = 0.451 and M = 3.83 ± 0.80; $d$ = 0.419), F-SC (M = 4.11 ± 0.81; $d$ = 0.364 and M = 4.26 ± 0.74; $d$ = 0.549)

and P-SC (M = 3.47 ± 0.72; *d* = 0.522 and M = 3.57 ± 0.77; *d* = 0.596), compared to not practising any physical activity or sport.

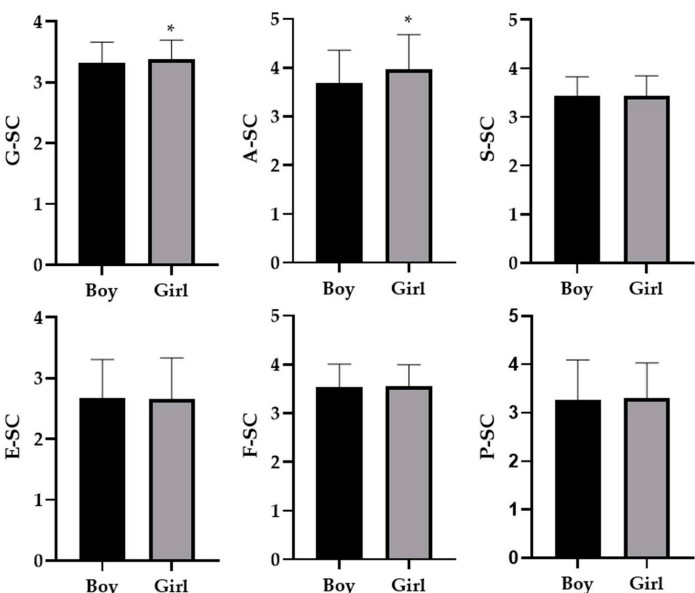

**Figure 3.** Gender-based differences in self-concept during COVID-19 lockdown. Note 1. Self-concept (SC); general self-concept (G-SC); academic self-concept (A-SC); social self-concept (S-SC); emotional self-concept (E-SC); family self-concept (F-SC); physical self-concept (P-SC). Note 2. $p \leq 0.05$ (*).

**Table 4.** Practice of physical or sport activities and self-concept before the COVID-19 lockdown.

| Variable | Sport | Mean | SD | F | Sig. | ES (*d*) | IC 95% |
|---|---|---|---|---|---|---|---|
| G-SC | NS | 3.30 | 0.49 | 5.826 | $p \leq 0.05$ [a,b] | 0.554 [a] <br> 0.574 [b] | [0.300; 0.808] [a] <br> [0.278; 0.871] [b] |
| | CIS | 3.35 | 0.35 | | | | |
| | NCIS | 3.58 | 0.52 | | | | |
| | CTS | 3.59 | 0.53 | | | | |
| | NCTS | 3.46 | 0.52 | | | | |
| A-SC | NS | 2.73 | 0.76 | 7.492 | $p \leq 0.05$ [a] | 0.905 [a] | [0.644; 1.167] [a] |
| | CIS | 3.16 | 0.88 | | | | |
| | NCIS | 3.47 | 0.87 | | | | |
| | CTS | 3.14 | 0.98 | | | | |
| | NCTS | 3.21 | 0.74 | | | | |
| S-SC | NS | 3.50 | 0.78 | 3.684 | $p \leq 0.05$ [a,b] | 0.451 [a] <br> 0.419 [b] | [0.149; 0.737] [a] <br> [0.125; 0.713] [b] |
| | CIS | 3.58 | 0.62 | | | | |
| | NCIS | 3.83 | 0.68 | | | | |
| | CTS | 3.83 | 0.80 | | | | |
| | NCTS | 3.69 | 0.72 | | | | |
| E-SC | NS | 2.96 | 0.77 | 1.552 | $p \geq 0.05$ | NP | NP |
| | CIS | 3.31 | 0.81 | | | | |
| | NCIS | 3.01 | 0.82 | | | | |
| | CTS | 3.17 | 0.77 | | | | |
| | NCTS | 2.88 | 0.63 | | | | |

**Table 4.** *Cont.*

| Variable | Sport | Mean | SD | F | Sig. | ES (*d*) | IC 95% |
|---|---|---|---|---|---|---|---|
| F-SC | NS | 3.80 | 0.89 | 4.210 | $p \leq 0.05$ [a,b] | 0.364 [a] 0.549 [b] | [0.113; 0.615] [a] [0.253; 0.845] [b] |
| | CIS | 3.83 | 0.72 | | | | |
| | NCIS | 4.11 | 0.81 | | | | |
| | CTS | 4.26 | 0.74 | | | | |
| | NCTS | 4.06 | 0.81 | | | | |
| P-SC | NS | 3.10 | 0.80 | 5.562 | $p \leq 0.05$ [a,b] | 0.522 [a] 0.596 [b] | [0.299; 0.806] [a] [0.299; 0.892] [b] |
| | CIS | 3.30 | 0.51 | | | | |
| | NCIS | 3.47 | 0.72 | | | | |
| | CTS | 3.57 | 0.77 | | | | |
| | NCTS | 3.46 | 0.86 | | | | |

Note 1. General self-concept (G-SC); academic self-concept (A-SC); social self-concept (S-SC); emotional self-concept (E-SC); family self-concept (F-SC); physical self-concept (P-SC). Note 2. No sport(NS); non-contact individual sport (NCIS); contact individual sport (CIS); non-contact team sport (NCTS); contact team sport (CTS). Note 3. Differences between NP and NCIS ([a]); differences between NP and CTS ([b]).

Figure 4 compares groups of physically active and inactive individuals before the lockdown. Being physically active before lockdown was associated with higher levels of G-SC, A-SC, S-SC, F-SC and P-SC. However, no statistically significant differences ($p \geq 0.05$) were found for the E-SC dimension.

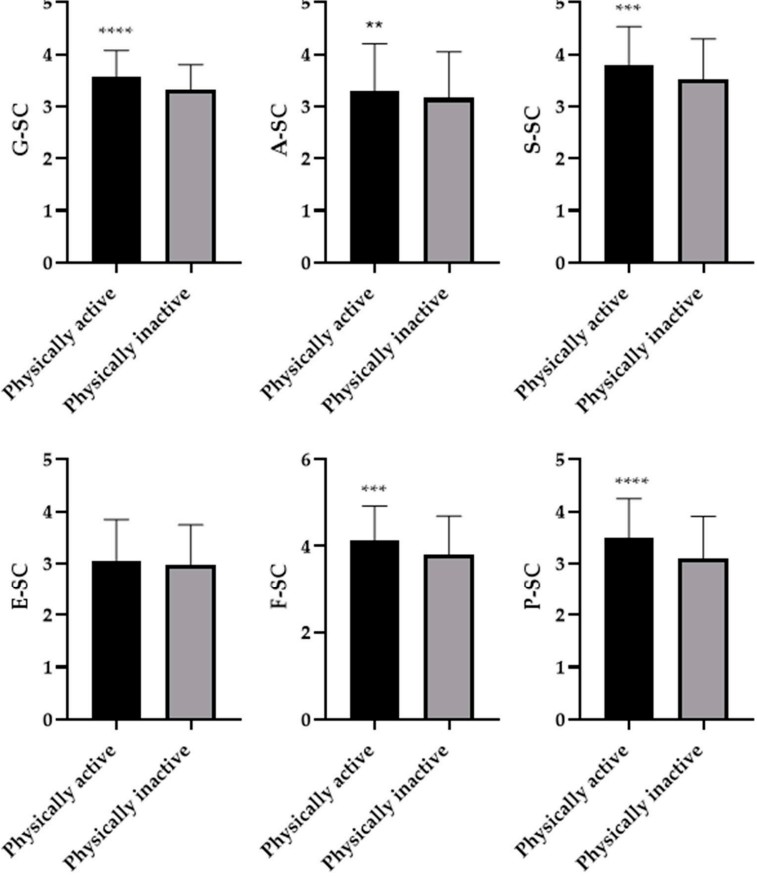

**Figure 4.** Self-concept and dimensions concerning physically active and inactive adolescents before the lockdown. Note 1. Self-concept (SC); general self-concept (G-SC); academic self-concept (A-SC); social self-concept (S-SC); emotional self-concept (E-SC); family self-concept (F-SC); physical self-concept (P-SC). Note 2. $p \leq 0.01$ (**); $p \leq 0.001$ (***); $p \leq 0.0001$ (****).

For the association between the practice of physical or sport activities and self-concept during the lockdown (Table 5), significant differences were found ($p \leq 0.05$). The subjects who performed NCIS activities (M = 3.48 ± 0.26; $d$ = 0.827) were those who had higher values in G-SC compared to non-realised physical activity (M = 3.22 ± 0.66). No physical activity was associated with a greater A-SC (M = 4.17 ± 0.58; $d$ = 0.697). Adolescents who practiced NCIS (M = 3.13 ± 0.60; $d$ = 1.194) or CTS (M = 2.81 ± 0.57; $d$ = 0.698) had higher values of E-SC than those not practicing any physical activity or sport (M = 2.37 ± 0.67). Individuals who practised NCIS (M = 3.52 ± 0.62; $d$ = 0.922) and CTS (M = 3.49 ± 0.78; $d$ = 0.907) presented the highest values of P-SC in comparison to those who did not practise any activities (M = 2.88 ± 0.76).

**Table 5.** Practice of physical or sports activities and self-concept during COVID-19 lockdown.

| Variable | Sport | Mean | SD | F | Sig. | ES (*d*) | IC 95% |
|---|---|---|---|---|---|---|---|
| G-SC | NS | 3.22 | 0.36 | 4.209 | $p \leq 0.05$ [a] | 0.827 [a] | [0.394; 1.259] [a] |
|  | CIS | 3.44 | 0.12 |  |  |  |  |
|  | NCIS | 3.48 | 0.26 |  |  |  |  |
|  | CTS | 3.36 | 0.31 |  |  |  |  |
|  | NCTS | 3.33 | 0.28 |  |  |  |  |
| A-SC | NS | 4.17 | 0.58 | 3.583 | $p \leq 0.05$ [a] | 0.697 [a] | [0.269; 1.125] [a] |
|  | CIS | 3.66 | 0.44 |  |  |  |  |
|  | NCIS | 3.64 | 0.69 |  |  |  |  |
|  | CTS | 3.74 | 0.67 |  |  |  |  |
|  | NCTS | 3.69 | 0.78 |  |  |  |  |
| S-SC | NS | 3.34 | 0.46 | 1.531 | $p \geq 0.05$ | NP | NP |
|  | CIS | 3.38 | 0.22 |  |  |  |  |
|  | NCIS | 3.54 | 0.35 |  |  |  |  |
|  | CTS | 3.44 | 0.39 |  |  |  |  |
|  | NCTS | 3.35 | 0.34 |  |  |  |  |
| E-SC | NS | 2.37 | 0.67 | 1.125 | $p \leq 0.05$ [a,b] | 1.194 [a] 0.698 [b] | [0.743; 1.645] [a] [0.231; 1.164] [b] |
|  | CIS | 2.57 | 0.58 |  |  |  |  |
|  | NCIS | 3.13 | 0.60 |  |  |  |  |
|  | CTS | 2.81 | 0.57 |  |  |  |  |
|  | NCTS | 2.63 | 0.75 |  |  |  |  |
| F-SC | NS | 3.51 | 0.47 | 0.483 | $p \geq 0.05$ | NP | NP |
|  | CIS | 3.47 | 0.55 |  |  |  |  |
|  | NCIS | 3.62 | 0.38 |  |  |  |  |
|  | CTS | 3.49 | 0.51 |  |  |  |  |
|  | NCTS | 3.53 | 0.37 |  |  |  |  |
| P-SC | NS | 2.88 | 0.76 | 5.714 | $p \leq 0.05$ [a,b] | 0.922 [a] 0.907 [b] | [0.485; 1.359] [a] [0.432; 1.382] [b] |
|  | CIS | 3.55 | 0.38 |  |  |  |  |
|  | NCIS | 3.52 | 0.62 |  |  |  |  |
|  | CTS | 3.49 | 0.78 |  |  |  |  |
|  | NCTS | 3.33 | 0.77 |  |  |  |  |

Note 1. General self-concept (G-SC); academic self-concept (A-SC); social self-concept (S-SC); emotional self-concept (E-SC); family self-concept (F-SC); physical self-concept (P-SC). Note 2. No Sport (NS); non-contact individual sport (NCIS); contact individual sport (CIS); non-contact team sport (NCTS); contact team sport (CTS). Note 3. Differences between NP and NCIS ([a]); differences between NP and CTS ([b]).

Figure 5 compares groups of physically active and inactive adolescents during the lockdown. Being physically active during lockdown was associated with higher levels of G-SC, A-SC, S-SC and P-SC. However, no statistically significant differences ($p \geq 0.05$) were found in the E-SC and F-SC dimensions.

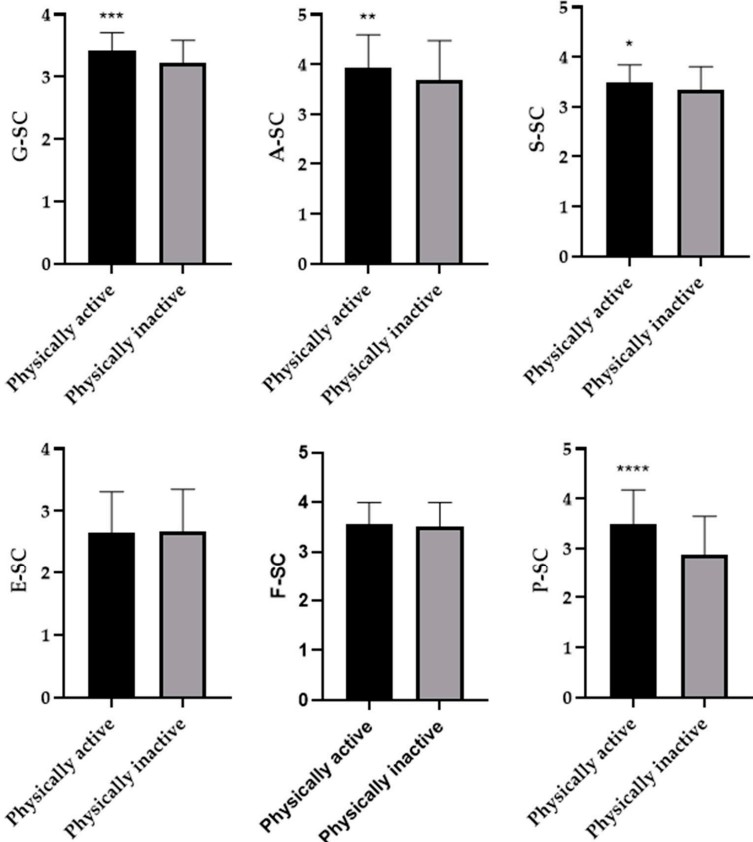

**Figure 5.** Self-concept and dimensions concerning physically active and inactive adolescents before the lockdown. Note 1. Self-concept (SC); general self-concept (G-SC); academic self-concept (A-SC); social self-concept (S-SC); emotional self-concept (E-SC); family self-concept (F-SC); physical self-concept (P-SC). Note 2. $p \leq 0.05$ (*); $p \leq 0.01$ (**); $p \leq 0.001$ (***); $p \leq 0.0001$ (****).

## 4. Discussion

This study aims to establish the association betweenself-concept, gender and the practice of physical activities, as well as to estimate the magnitude of their differences, in adolescents before and during the COVID-19 lockdown. There area wide variety of studies that addressed this field before the COVID-19 lockdown [19,37–39], yet evidence evaluating adolescents' self-concept during the COVID-19 pandemic was not found. The strength of this research relies on comparing adolescents' self-concept before and during the COVID-19 pandemic, as well as estimating differences based on gender and the practice of physical activity. This is a crucial area of research since self-concept is a pivotal construct in adolescence [40,41], given its association with quality and life satisfaction, and because it is influenced by social, physical, emotional, academic and family features [40,42,43].

Adolescents' general and social self-concept were higher before the lockdown. However, the emotional detriment during the lockdown caused lower values of self-concept in comparison to before the isolation. In this sense, studies such as Martínez-Marín, Martínez and Paterna [44] evinced that self-concept predicted emotional intelligence, both genders being associated with clarity and emotional repair. In line with this, an appropriate emotional state is associated with a better self-perception, self-esteem and self-rate in youth [45].

Emotional self-concept was higher before the lockdown, and it was correlated to contact with family and peers. Findings reveal that as adolescents' knowledge increases, so positive/negative emotion regulation and psychological adjustment are more efficient [46]. Hence, participants' family self-concept was higher before the lockdown, since adolescents who reported a high family functioning had appropriate values of self-concept [20].

However, during the COVID-19 lockdown there was a negative association between physical and emotional components, as well as higher values of academic self-concept. Staying at home, with resources enough to maintain the normality, meant that adolescents had a higher academic self-concept, which led to higher academic performance [47]. In this sense, perceived competences are demonstrated to buffer the association between academic self-concept and goal-oriented motivation [48].

As regards gender-based differences, females had higher values of academic self-concept in both moments. Moreover, males' academic self-concept was negatively correlated to emotional self-concept, yet family self-concept was positively associated with their academic self-concept during the lockdown. Furthermore, boys had higher levels in physical and emotional self-concept before the lockdown, while girls' emotional self-concept was correlated to support and proximity of family and peers. Males usually have higher emotional and physical self-concept, whereas females tend to have higher academic and family self-concept [19,41,46,48–51]. Age intensifies this difference, and some aspects were revealed, such as peers' relations, academic performance or motor-perceptive skills consolidation [21,52]. On the whole, girls' general self-concept was higher during the lockdown.

The gender-based difference in terms of the dimensions of self-concept causes some differences while choosing a physical activitie, non-contact individual sports and contact team sports being chosen more frequently by adolescents, which are associated with identity acceptance [19,32,53]. However, girls usually are less involved in these types of activity, yet benefits and enhancement on self-concept are very similar if both genders take part in physical activities in the same way [19,54]. Following on from the foregoing, practising no physical activities led to low levels of self-concept before the lockdown; indeed, there was a positive association between self-concept dimensions and the level of physical activity, and self-concept was negatively associated with sedentary behaviour [24]. In line with this, organised physical activity fostered a higher meeting of needs in the youngest [37].

Nevertheless, practising non-contact individual sports was associated with a higher academic self-concept in adolescents, as opposed to those who practised no physical activity. The practice of physical activity and the decrease in sedentary behaviour were generally correlated to academic performance and cognitive development [55–58], so that these factors are positively associated with a higher academic self-concept [59,60]. Likewise, adolescents who practised either non-contact individual sports or contact team sports had higher social, family and physical self-concept in comparison to those who did not practise any physical activity. Generally, it has been demonstrated that the benefits of practising a moderate physical activity in adolescence promote the development of self-concept and help to optimise health and welfare in the long term [61].

During the COVID-19 pandemic, some studies exposed that the practice of physical activity was a useful tool to cope with this situation, as well as to improve people's health and their ability to enhance their self-growth [62,63]. Lack of regular practice of physical activity within this period led to harmful consequences [64–70]. Nevertheless, some countries allowed the off-site practice of physical activity such as biking as a mean of transport, providing that people maintained social distance, which reported a lower health detriment due to both physical activity and social distance [71]. In this investigation, there was a positive association between the practice of individual physical activities and general/physical self-concept in adolescents. Thus, adolescents with higher emotional strength were the ones who expressed practising physical activity. In line with this, psychological consequences during COVID-19 pandemic have also been studied, so the lockdown resulted in harmful psychological consequences, although physical activity was associated with the prevention of psychology disruptions since it helps to control anxiety, decrease psychosomatic symptoms and enhance self-esteem [72–74]. However, adolescents who did not practise any physical activity during the isolation had higher values

in academic self-concept. Time expenditure during this lockdown is focused on doing daily tasks such as assignments, practising physical activity and using IT devices [46].

The return to habitual practice of physical activity and its link with COVID-19 is not established yet. Experts are working on protocols that prevent the virus spreading so that people may practise physical or sports activities during a lockdown due to their physical, social, psychological and cognitive benefits [75–78]. This is a crucial matter because this is not the only global pandemic we are supposedly suffering in future. Lastly, it must be highlighted that the World Health Organization [79] states that an active life enhances self-perception [19].

The present study has some limitations. It is important to emphasise some caveats to the findings of this research for generalisation to other populations such as the population size. Furthermore, the descriptive design did not allow for establishing causal association (test-retest), so the results obtained ought to be interpreted cautiously. To the knowledge of the authors of this study, another limitation has been the lack of solid research that worked on the study variables during the COVID-19 pandemic. Consequently, it would be interesting to widen the sample by including more adolescents across the globe so that comparisons may be possible. Moreover, it would be interesting to evaluate the population once the COVID-19 pandemic finishes, establishing associations within the entire period. Lastly, from another perspective and as a practical application, it has been demonstrated that a cognitive-behaviour intervention and the practice of physical activities may help to obtain higher family, emotional and physical self-concept, as well as a decrease in psychopathological symptoms, in harmful situations in which psychosocial problems appear and there is a detriment of mental health [22,30,80,81].

## 5. Conclusions

The aim of the study was to assess the level of self-concept of adolescents before and during the lockdown, as regards gender and physical activity. Therefore, it is concluded that adolescents' self-rate and self-concept before the lockdown were higher than during the COVID-19 pandemic, which is positively associated with emotional self-concept. Moreover, family and peers were essential factors for an appropriate emotional self-concept development before and during the lockdown. Furthermore, academic self-concept during the lockdown was higher than before it.

It may be highlighted that females' self-concept was higher during the isolation than the males' one. Specifically, girls had a higher academic self-concept in both times, although boys' academic self-concept was influenced by their emotional management and their relationships with family and peers. Even though males' physical and emotional self-concept was higher than the females' ones before the lockdown, these differences did not last during the lockdown.

As regards physical activity, people who did not practise any physical or sports activity had a lower self-concept before the lockdown. During the isolation, practising physical activity before it was associated with higher academic, social, family and physical self-concept, as opposed to those who practised no physical activity. However, academic self-concept was positively associated with not practising physical activity. Within the lockdown, adolescents who practised non-contact individual sports had higher general and emotional self-concept. Lastly, before the lockdown, adolescents who practised physical activity reported higher physical self-concept.

**Author Contributions:** Conceptualisation, G.G.-V. and F.Z.-O.; methodology, G.G.-V., F.Z.-O. and D.L.-P.; software, G.G.-V. and F.Z.-O.; formal analysis, G.G.-V., F.Z.-O. and J.C.-P.; investigation, G.G.-V., F.Z.-O., D.L.-P. and J.C.-P.; data curation, G.G.-V.; writing—original draft preparation, G.G.-V. and D.L.-P.; writing—review and editing, G.G.-V., F.Z.-O., D.L.-P., G.B. and J.C.-P.; visualisation, G.G.-V., G.B., W.R.G. and F.Z.-O.; supervision, G.G.-V., G.B., W.R.G. and F.Z.-O. All authors have read and agreed to the published version of the manuscript.

**Funding:** This research received no external funding.

**Conflicts of Interest:** The authors declare no conflict of interest.

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
