# Peer review of "Analysis of Self-Concept in Adolescents before and during COVID-19 Lockdown: Differences by Gender and Sports Activity"

_sustainability, doi:10.3390/su12187792_

Round 1
Reviewer 1 Report
Dear authors,
Work has improved with the changes made. I recommend some fixes:
-Line 78 “An ex post facto (non-experimental) study was conducted, with a descriptive, comparative, correlational and cross-sectional design, and a single measurement on a single group was taken”. Do I take only one measurement? How do you justify that the work is scientifically sound? Justify if the sample is representative?
-Line 83. Mention that “Convenience sampling was used to select participants, in which adolescents were asked to participate before and during the lockdown, so different samples were evaluated at two different times”. Explain how you knew the phenomenon of the global health crisis would occur, that is, how do you compare the data before and during the pandemic?
- It is admissible such a high number of self-citations. Six have been counted. Remove the references: 22, 32 and 41.
-In section 3. Results (lines 142-261). You must make an effort to make it fluid for the reader; in addition to summarizing the final concepts of the work.
-Add the limitations of the study and the methodology used.
Author Response
Dear Reviewer,
I attach the required additions.
Thank you!

Reviewer 2 Report
I have responded to the authors' responses to my initial comments in bold after your responses. I have made a number of suggestions at the end which may help the article with consistency in the use of terms.
I have also attached a document which indicates the level of editing I think is required.
REVIEWER 2 Comment 1.
The introduction should conclude with a clear statement of the research question/s and/or aims for this study. Therefore, the following statement may be a clearer statement of the aims of the study...
"This study aims to examine the level of adolescents' self-concept before lockdown and during lockdown, for boys and girls, and for those who are physically active and those who are not.
Response 1.
Thank you very much for your comments. In the last paragraph of the introduction, the research questions were included. The aim has been modified on the basis of their recommendations. Thank you for your contribution.
However, the actual research questions need to be stated more clearly. Suggestions are made in attached document.
Comment 2.
There is no literature review. The literature referred to in the discussion section would have been useful to set the context for the rationale and design of this study; and to justify this study as important for contributing to the field.
Response 2.
Thank you very much for your comments. We consider that the introduction should be a section in which the problems and the main characteristics of the evaluated constructs are established, being the discussion one of the most important sections (reason why it has been discussed with more studies). However, thanks to their comments, the introduction has been improved by including authors used in the discussion.
OK
Comment 3.
The sample/s are not well described and are confusing. It appears that students were recruited (agreed to participate) and came from a single high school programme. Most of them (366) responded in the no-lockdown period and 164 responded during the lockdown period. Therefore, you have a subset of students who are common to both groups. Given the nature of the analyses they are actually separate groups of individuals. Is this correct? At what point were data of 37 adolescent discarded – the lockdown sample or the no-lockdown group? You don't treat the sample as a single group (where you would have a repeated-measures design, for some some of them anyway). However, if you want to compare the responses of the two groups, then it is important to establish that they are equivalent in all respects except for the lockdown status. E.g. for each sample it is necessary to know the gender balance, and age range. It is inappropriate to simply report the overall gender balance and age range for the sample (the two group combined).
Response 3.
Thank you very much for your comments. The section "Design and participants" has been modified.
In order not to confuse readers, the term "quantitative" has been removed and it has been added that this is non-experimental research.
I don’t think you need to ‘label the design’. I suggest removing this all together. The study is simply a comparison of the self-concept of two groups of adolescents, one pre-lockdown and one during lockdown.
It has been specified that this is not a single group, as two different groups have been evaluated at two different times. If it is true, that methodologically we have been informed and this type of sampling is called "in a single group at a single time" (non- experimental). In this way, it is clear that there is no common subset.
You don’t need to describe the whole group – this is irrelevant. It is good that you have described the two groups being compared. They are not exactly the same (e.g. the lockdown group is smaller, younger and has more females). Need to state this and decide if they are similar enough not to be a problem.
It has been clarified at the time that the 37 adolescents were excluded and it was made clear that they are equivalent groups except in the situation of confinement. In this sense, the age range and gender have been added according to the groups.
OK I suggest you move this statement to the data analysis section.
Comment 4.
An alternative design would be to use the 136 students who responded at both times? Then you would have the same sample responding to the %-FR on the two occasions. This would have been much tidier in terms of investigating the effect of lockdown on self-concept, by gender and physical activity.
Response 4.
Thank you very much for this recommendation. As we have previously clarified, this is a non-experimental research since the subjects themselves did not respond and were not evaluated over time. This study aims to make a comparison of the two groups. We appreciate the initiative to carry out a pre-experimental study (no control group in this case). Thanks again for the suggestions.
OK, it is much clearer now that separate groups were used.
Comment 5.
While the alpha for the AF-5 for the total scale is .80, it would also be useful to site the alpha for each sub-scale as these are used as dependent variables as well.
Response 5.
Thank you for your comment. The reliability of the subscales of the self-concept were added.
OK. This is good. Just clarify that the alphas you give relate to your data.
Comment 6.
Establishing the invariance of the two groups being compared is appropriate, but it needs to be followed through with. One of the assumptions of the t-test and F-test is that the variances are equivalent. However, this was not the case for the analyses in Table 1.
Why correlate the mean scores on each variable? This would inflate the correlation coefficients. It would be more appropriate to calculate correlations on individual responses and be a more accurate estimate of the relationship. I assume that TE (d) (used in tables) means effect size?
I question the use of correlations to interpret the differences between groups on the SC sub-scales. The correlations could be more usefully used to establish the inter- relationship between the sub-scales and each with the overall scale.
Response 6.
Thank you for your comments. The article has been improved thanks to your comments.
TE" actually refers to the effect size. This has been replaced by "ES".
Good
We understand and appreciate that you may question the use of correlations and agree that one of the most interesting aspects is to make that comparison with the global scale. However, we also consider it appropriate to show readers the correlation rates between the subscales in different situations. These indicators help us to draw conclusions about which aspects directly or indirectly affect these dimensions (academic, social, family, emotional and physical) before and during confinement, in order to show a practical application of the study (future perspectives).
I accept that correlations might be useful. I was questioning the use of means rather than individual scores.
Comment 7.
The Levene test (for homogeneity) between the two groups are all statistically significant, indicating that they are not similar. This does not meet one of the assumptions of a t-test. Suggest you use ES instead of TE of d. There is a confusing mix of symbols used for ES.
The findings have not been presented very clearly. E.g. the essential points from Table 1 are (i) overall self-concept is greater in no-lockdown than in lockdown; (ii) same pattern for S-SC, E-SC and F-SC; opposite for A-SC and not significant for P-SC. In some respects, the correlational analyses are irrelevant (except for establishing that they are all positively related to G-SC).
A two-way ANOVA would have been more appropriate to use than separate t-tests for the lockdown and no-lockdown periods.
Table 3. It might have been sufficient to compare physically active versus inactive groups of students, as the various types of physical activities is distracting and not the findings are not compelling.
Inactive group has lower G-SC, S-SC, F-SC, P-SC; and higher A-SC.
Response 7.
Thank you for your comments.
ES" has been changed to "TE.
OK
Thanks to your comment. The results have been clearly presented. Although you give us a synthetic example (synthesis) of the main results, we insist that we consider it appropriate to relate it to the correlation indices, as the statistical index "r" can help to draw conclusions about the differences in means, due to these direct or indirect associations. Furthermore, based on the comments of the reviewer 1, the figures of each of the relationships have been added.
The addition of graphs has been useful.
We agree and appreciate the comment that it would have been more appropriate to conduct two-way ANOVA, however we have conducted the analyses and the results are the same. We have simply thought that it is more appropriate to interpret the effect of relationship with the d of Cohen (Cohen 1988) than with the statistical index eta squared.
You need to explain this then. There is no reason you cannot use Cohen’s d with ANOVA.
I think you need to be selective about what information you present in the tables and not simply use a printout of results from the software program. I made comments about this previously.
It would be useful to refer to the effect sizes for significant differences as these reveal very useful information about the relative size/importance of the difference. E.g. ES of .900 is substantially higher than .600
In relation to your comments and to the reviewer 1, statistical graphs made with the programme "GraphPad Prism 8 (GraphPad Prism Software Inc., San Diego, CA, USA)" have been included. In this case, the tables in which the classification was based on the type of sport modality have been maintained, but the analysis of the groups based on "physically active" and "physically inactive" has been included, being this represented with a figure.
You need to explain this.
Comment 8.
Discussion. Much of this material should be presented in a literature review to support a rationale for this particular study. This would make the discussion more succinct. The relationship of the findings from this study and the literature are noted.
Response 8.
Thank you very much for your comments. We appreciate your feedback on the discussion. In relation to your review, relevant information was included in the introduction to make the discussion part more meaningful.
OK
Comment 9.
Conclusions. These are appropriate
Response 9.
Thank you very much.
Suggestions:
Title: Analysis of self-concept in adolescents before and during
COVID-19 lockdown: Differences by gender and sports activity
Use the term ‘adolescent’ throughout instead of youth or student or minor
‘Activity’ not ‘activeness’
‘Lockdown’ not ‘confinement’
Why use ‘Female/Girl’ under the tables? Use one or the other. Same with ‘boys/makes’
Clearer titles for Figures and Tables
E.g. Figure 1 requires a note that ‘Yes’= lockdown; ‘No’ = no lockdown
Table 5 No Practitioner (NP). ‘Practitioner’ is not the appropriate word to use here. This should be ‘Not active’ (NA) or ‘No sport’ (NS)
‘physical-sports activities’ – this could be referred to ‘sports’ or ‘physical activities’

Author Response

(The authors gave the same response as above.)

Round 2
Reviewer 1 Report
Dear authors,
The work has improved with the changes made, although I recommend some fixes:
-The introduction section should include the structure of the rest of the manuscript.
- A conceptual framework should be included as section 2 in which the most relevant terms of the study are described in their context.
-Develop discussions based on these defined concepts.
-In the research questions the question marks are missing.
- When you talk about "boys and girls", do you mean teenagers?
-Tables 1,2,3 could be improved.
Author Response
Dear Reviewer,
I attach the letter of changes.
Thank you!

This manuscript is a resubmission of an earlier submission. The following is a list of the peer review reports and author responses from that submission.
Round 1
Reviewer 1 Report
Dear authors,
The submitted manuscript has very good intentions and although it is a topic of interest to the scientific community, there are more consistent works and I observe some weaknesses in its study. I comment on them below:
- It is recommended to review the title of the manuscript. It must be concise and concrete and summarize the subject of study.
- Synthesize the abstract and highlight results and conclusions in a concise way.
- Delve into the introduction that presents forcefulness on the background of the subject in question.
- It is recommended to include a table and graph to the results.
The results are not consistent. Review the tables.
- Expand the conclusions and relate them to the main objective.
- Connect the conclusions in relation to the results obtained so that they present greater forcefulness. Indicate how the results obtained are related or not to the literature and the opinion of prominent authors on the subject.
Reviewer 2 Report
This is potentially a very interesting and useful piece of research in the current global environment. However, I found there were a number of issues that need to be addressed because they were either unclear or inappropriate.
1. Introduction
The introduction should conclude with a clear statement of the research question/s and/or aims for this study. Therefore, the following statement may be a clearer statement of the aims of the study...
"This study aims to examine the level of adolescents' self-concept before lockdown and during lockdown, for boys and girls, and for those who are physically active and those who are not.
2. Literature Review
There is no literature review. The literature referred to in the discussion section would have been useful to set the context for the rationale and design of this study; and to justify this study as important for contributing to the field.
3. Methods.
The sample/s are not well described and are confusing. It appears that students were recruited (agreed to participate) and came from a single high school programme. Most of them (366) responded in the no-lockdown period and 164 responded during the lockdown period. Therefore, you have a subset of students who are common to both groups. Given the nature of the analyses they are actually separate groups of individuals. Is this correct? At what point were data of 37 adolescent discarded – the lockdown sample or the no-lockdown group? You don't treat the sample as a single group (where you would have a repeated-measures design, for some some of them anyway). However, if you want to compare the responses of the two groups, then it is important to establish that they are equivalent in all respects except for the lockdown status. E.g. for each sample it is necessary to know the gender balance, and age range. It is inappropriate to simply report the overall gender balance and age range for the sample (the two group combined).
Design – It is better to not try and 'label' the design as it is very confusing and incorrect as stated. The design is better described as a "quantitative" (although it is not clear what overlap there was between the two groups. There are two groups (not a single sample); a single measure (AF-5) but a total scale and several sub scales
No need to label the questionnaire as 'post hoc'. When was it administered?
An alternative design would be to use the 136 students who responded at both times? Then you would have the same sample responding to the %-FR on the two occasions. This would have been much tidier in terms of investigating the effect of lockdown on self-concept, by gender and physical activity.
While the alpha for the AF-5 for the total scale is .80, it would also be useful to site the alpha for each sub-scale as these are used as dependent variables as well.
Why correlate the mean scores on each variable? This would inflate the correlation coefficients. It would be more appropriate to calculate correlations on individual responses and be a more accurate estimate of the relationship. I assume that TE (d) (used in tabes) means effect size?
I question the use of correlations to interpret the differences between groups on the SC sub-scales. The correlations could be more usefully used to establish the inter-relationship between the sub-scales and each with the overall scale.
4. Results
The Levene test (for homogeneity) between the two groups are all statistically significant, indicating that they are not similar. This does not meet one of the assumptions of a t-test. Suggest you use ES instead of TE of d. There is a confusing mix of symbols used for ES.
The findings have not been presented very clearly. E.g. the essential points from Table 1 are (i) overall self-concept is greater in no-lockdown than in lockdown; (ii) same pattern for S-SC, E-SC and F-SC; opposite for A-SC and not significant for P-SC. In some respects, the correlational analyses are irrelevant (except for establishing that they are all positively related to G-SC).
A two-way ANOVA would have been more appropriate to use than separate t-tests for the lockdown and no-lockdown periods.
Table 3. It might have been sufficient to compare physically active versus inactive groups of students, as the various types of physical activities is distracting and not the findings are not compelling.
Inactive group has lower G-SC, S-SC, F-SC, P-SC; and higher A-SC.
5. Discussion
Much of this material should be presented in a literature review to support a rationale for this particular study. This would make the discussion more succinct. The relationship of the findings from this study and the literature are noted.
6. Conclusions
These are appropriate.